# Single-cell sequencing of genomic DNA resolves sub-clonal heterogeneity in a melanoma cell line

Enrique I. Velazquez-Villarreal[1,5 ✉], Shamoni Maheshwari[2,5], Jon Sorenson[2], Ian T. Fiddes[2], Vijay Kumar[2], Yifeng Yin[2], Michelle G. Webb[1], Claudia Catalanotti[2], Mira Grigorova[3], Paul A. Edwards [3,4], John D. Carpten [1] & David W. Craig[1 ✉]

We performed shallow single-cell sequencing of genomic DNA across 1475 cells from a cell-line, COLO829, to resolve overall complexity and clonality. This melanoma tumor-line has been previously characterized by multiple technologies and is a benchmark for evaluating somatic alterations. In some of these studies, COLO829 has shown conflicting and/or indeterminate copy number and, thus, single-cell sequencing provides a tool for gaining insight. Following shallow single-cell sequencing, we first identified at least four major sub-clones by discriminant analysis of principal components of single-cell copy number data. Based on clustering, break-point and loss of heterozygosity analysis of aggregated data from sub-clones, we identified distinct hallmark events that were validated within bulk sequencing and spectral karyotyping. In summary, COLO829 exhibits a classical Dutrillaux's monosomic/ trisomic pattern of karyotype evolution with endoreduplication, where consistent sub-clones emerge from the loss/gain of abnormal chromosomes. Overall, our results demonstrate how shallow copy number profiling can uncover hidden biological insights.

---

[1] Department of Translational Genomics, Keck School of Medicine of University of Southern California, Los Angeles, CA, USA. [2] 10X Genomics, Pleasanton, CA, USA. [3] Hutchison-MRC Research Centre, University of Cambridge, Cambridge, UK. [4] Cancer Research UK Cambridge Institute, Cambridge, UK. [5] These authors contributed equally: Enrique I. Velazquez-Villarreal, Shamoni Maheshwari. ✉email: eivelazq@usc.edu; davidwcr@usc.edu

Gaining a single-cell view of tumor heterogeneity is crucial for improving our understanding of tumor evolution and enabling future advances in cancer research. The standard paradigm is bulk sequencing of genomic DNA derived from millions of heterogeneous cells. In bulk sequencing, the ability to resolve sub-clonality is confounded relying on indirect inference, frequently resulting in an ensemble view dominated by the majority clone[1,2]. While bulk sequencing has provided major insights into tumor biology, lower throughput single-cell methods such as spectral karyotyping are often necessary to understand sub-clonal complexity and tumor evolution. Previously, methods for sequencing DNA of single-cells using next-generation sequencing approaches have often been laborious or limited to multiplexing hundreds of cells or nuclei[3–8]. In this study, we used newly emerging droplet-based shallow genome sequencing to simultaneously sequence 1475 single-cells from one of the most well-studied and well-characterized benchmark cell lines, COLO829[9], as a means to better understand limitations and insights gained by single-cell sequencing at shallow depth. We follow this analysis with a deep-dive, examining data by multiple technologies and multiple samples on COLO829, in order to better understand the resulting sub-clonality, its major hallmark features, and the underlying driving biology.

The melanoma COLO829 and germline COLO829-BL tumor/normal pair have been extensively analyzed using multiple methods and technologies, making it an ideal vehicle for new and emerging genomic technologies[1,2,10,11]. This line was one of the first tumor/normal pairs to be subjected to whole-genome sequencing, where Pleasance et al. identified several hallmark events including a homozygous 12 kb deletion in PTEN, BRAF 600V/E, and a CDK2NA 2 bp deletion. Previous studies using bulk sequencing of the tumor-line COLO829 have focused largely on developing tools and standardizations to improve copy number estimation and cancer characterization[2]. While a few of

the studies found cell line complexity inconsistent with the assumption of clonality and suggestive of multiple sub-clones, in general, most analyses presumed COLO829 to be a single clone. Of papers looking at copy number, Craig et al. observed differences among samples in chromosome 1p, and Gusnanto et al. found evidence for a mixture of clones but they were unable to resolve the individual components using bulk data and methods. Much of the work on this tumor-line highlighted major CNV hallmark events, as well as a series of inconsistent findings that point towards bulk sequencing methods being lossy and unable to resolve the complexity of COLO829[11].

Beyond the difficulty of resolving clonal mixtures, an additional challenge of bulk sequencing even in the context of a paired normal is that without single-cell resolution there are limited informatic options to resolve relative differences in read-depth to integer copy number states. At some point, most algorithms require assumptions, such as a diploid region or tumor purity, and the veracity of these assumptions shape overall accuracy. Even so, even with a uniform set of algorithms applied on the same cell line, variable results are observed across samples, suggesting that there may be differences with some sub-populations of cells impacting their expansion[11]. In this paper, we performed shallow single-cell sequencing of genomic DNA across 1475 cells from the same cell-line, COLO829, and show that it is in fact a complex mixture and identify key structural variants that contribute to its sub-clonal evolution.

## Results

**Copy number profiling at single-cell resolution**. We sequenced 1475 cells with 3044 billion $2 \times 100$ paired-end reads and conducted barcode-aware bioinformatic analysis using the Cell Ranger DNA Pipeline to call copy number profiles at single-cell resolution (Fig. 1b). The library had a median read duplication ratio of ~10% per cell, with on average 1,358,777 effective mapped

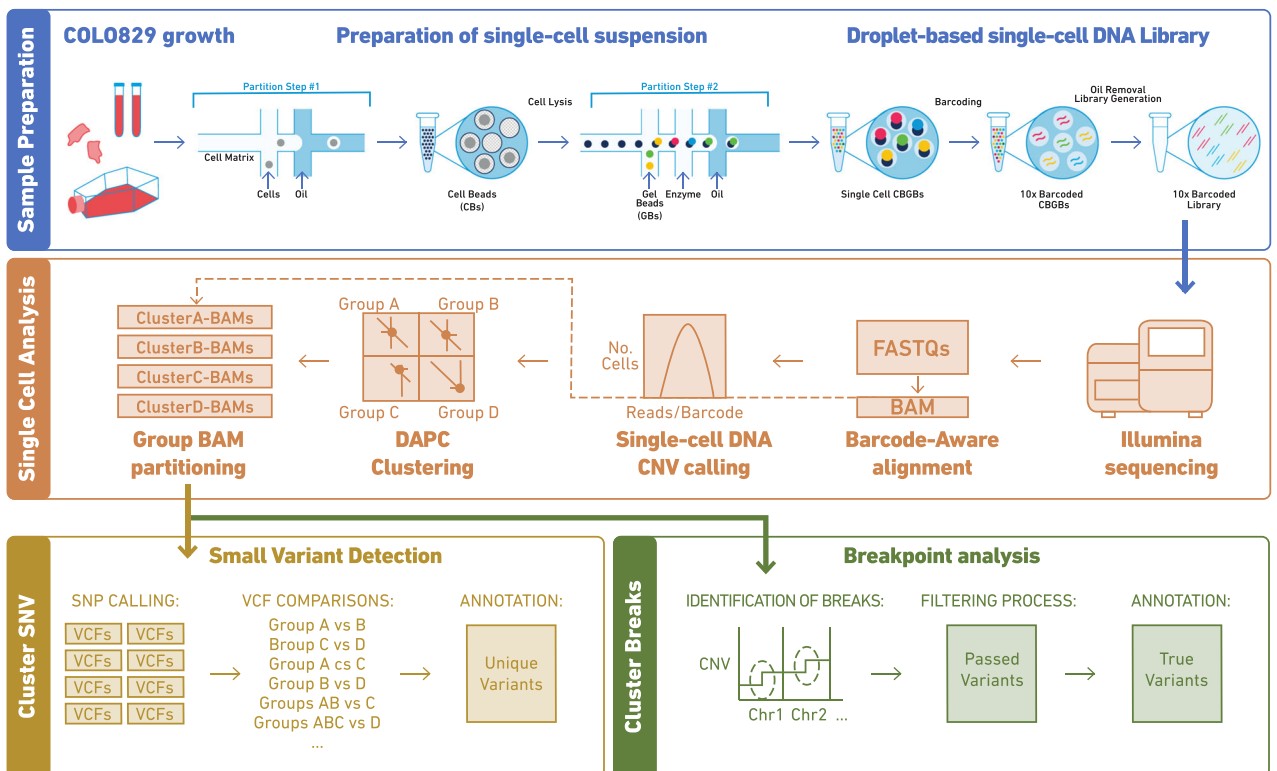

**Fig. 1 Workflow of the single-cells sample and data processing. a** Sample preparation of the chromium technology. **b** Single-cell analysis. Data processing from sample sequencing to BAM partitioning. **c** Data analysis including variant detection and breakpoint analyses.

deduplicated high-quality reads per cell and showed an average of 436 reads/Mb (Supplementary Fig. 2b). Notably, while the average genome-wide ploidy of single-cells was tightly distributed around a median ploidy of 3, the individual CNV profiles were different between cells (Supplementary Fig. 3). For example, cells at similar sequencing depths and average ploidies, cell #596 (448 reads/Mb and a mean ploidy of 3.03) and cell #415 (472 reads/Mb and a mean ploidy of 3.02) exhibit extensive copy number differences. At a 2 Mb resolution cell #596 has three distinct copy number states in chromosome 1 (3-2-4), two in chromosome 18 (2-3) and a single ploidy of 3 across chromosome 8; in contrast #415 exhibits two copy number states (2-4) in chromosome 1, while chromosome 18 is a single segment at copy number 2 and chromosome 8 is at a different copy number of 4 (Supplementary Fig. 3). Thus, despite uniform average genome-wide ploidies, single-cell resolution reveals cells having different copy number profiles in a single sample of COLO829 (Supplementary Fig. 4).

**COLO829 is composed of at least four major clusters**. We leveraged CNV events that were observed in this population to identify sub-clones and cluster single-cell CNV data. For this and all other downstream analysis, we excluded 6% of the cells that were flagged by the Cell Ranger DNA Pipeline as "noisy" and focused on the remaining 1373 single-cells. First, we filtered the raw CNV events by applying a size cutoff of >2 Mb and a quality cutoff of 15. Next, we derived a binary CNV event matrix tabulating the absence/presence of a CNV event across the 1373 single-cells (Methods: clustering of single-cell CNV data). This filtering resulted in 114 CNV events, with a majority <100 Mbp with ploidies of 2, 3 and 4 (Supplementary Fig. 5). Next, clustering was performed using the adegenet R package which uses Discriminant Analysis of Principal Components (DAPC), DAPC was used to identify groups of genetically related cells by constructing linear combinations of the original CNV events that have the largest between-group variance and the smallest within-group variance (Fig. 2a, Supplementary Fig. 6). For this analysis,

we chose a clustering solution of $k = 11$ guided by a BIC curve with the optimum ranging between 10 and 15 (Fig. 2b).

Plotting the single-cells on the coordinates of two primary DAPC axes revealed four distinct groups (A–D) of cells (Fig. 2c), with sub-structure within two of them (Fig. 2c). Visual inspection of single-cell CNV heatmap reveals striking clusters of chromosome-scale differences between cells (Fig. 3). Shown in Fig. 2, the DAPC clustering analysis indicates four major groups: Group A (653 cells), Group B (117 cells), Group C (43 cells), and Group D (560 cells). Group A and Group B are distinguished by a copy number of 3 on chromosome 8, whereas Group C and Group D have four copies of chromosome 8. Group B and Group C showed a loss of chromosome 1pter-1p22.3, chromosome 10p14-p11.22, and chromosome 18. Additional events are evident including on chromosomes 11 and 6, though we focus on the large-scale chromosomal differences evident between these four groups. A previous study, comparing four different cell-line samples through bulk sequencing analysis observed a similar 1p loss in one sample (referred to here as the TGen sample). To gain a better understanding of the relationship between these events, we created four group-level BAMs, one per major DAPC group to enable additional bulk-format analysis. We also utilized BAMs from the earlier bulk sequencing study by Craig et al.[2] to see if the events within these sub-clones were evident in the prior studies.

**Breakpoint analysis provides structural insights**. We next performed breakpoint analysis on the single-cell data by mapping anomalous read-pairs in order to identify the structural variants behind copy number changes. Specifically, we expect paired-read mapping to show how copy number segments were joined together. To ensure adequate power even in groups with low cell counts, we focused on anomalous read-pairs that aligned to regions within 2 Mb of the median breakpoint locations of the 114 shared CNV events identified in the above analysis. Breakpoint analysis on aggregated data showed DAPC group-specific anomalous read patterns that indicated clusters of breakpoints

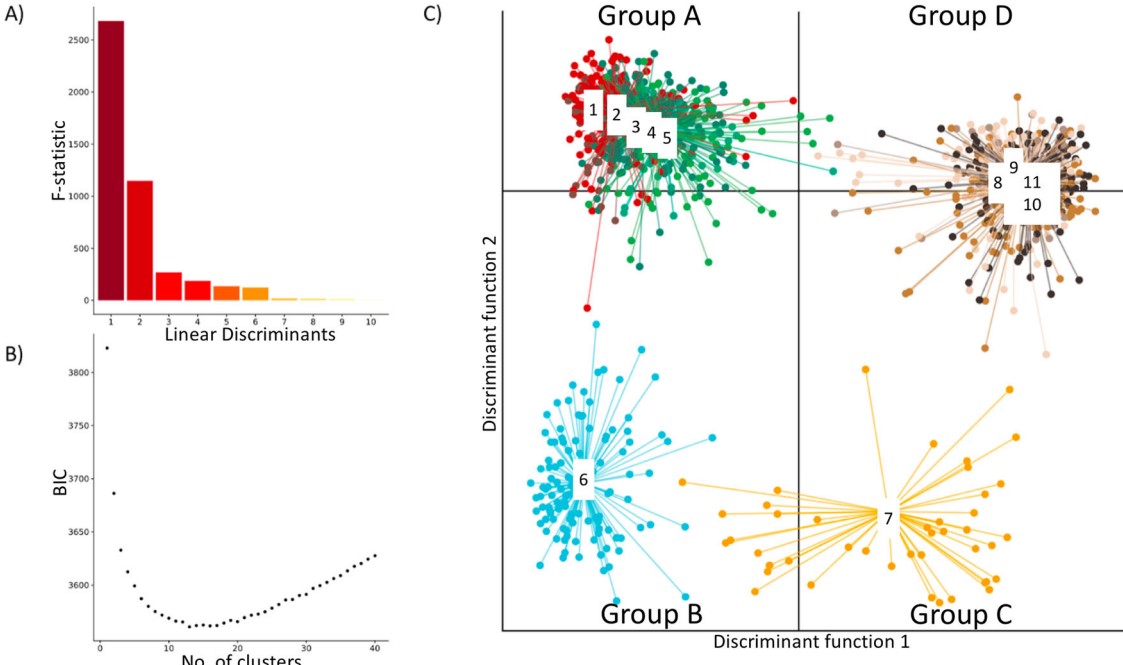

**Fig. 2 Clustering of COLO829 single-cells using Discriminant Analysis of Principal Components (DAPC). a** Bar plot of eigenvalues, which corresponds to the ratio of the variance between groups over the variance within groups for each discriminant function. **b** Inference of optimal cluster number using Bayesian Information Criterion (BIC). **c** Scatterplots showing the inference of population structure in 1373 cells using the first two principal components of the DAPC analysis. Individual dots represent single-cells and the color represents cluster assignment.

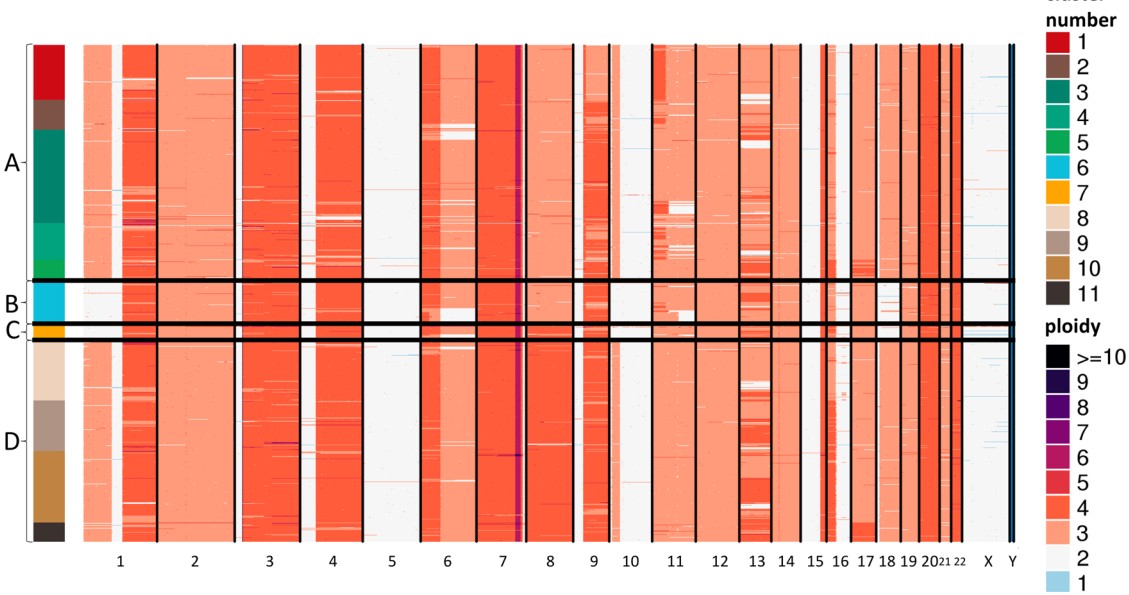

**Fig. 3 single-cell copy number heatmap of COLO829.** Heatmap showing hierarchical clustering of 1,373 single-cell CNV profiles at 2 Mb resolution. Each row depicts the whole genome of a single-cell, colors (Group A: clusters 1–5 [red, brown, greens], Group B: cluster 6 [blue], Group C: cluster 7 [yellow], Group D: clusters 8–11 [browns]) represent the called ploidy as specified by the legend on the right and rows are clustered by groups (Group A: 653 cells, Group B: 117 cells, Group C: 43 cells, Group D: 560). Hierarchical clustering was performed calculating the distance between each single-cell CNV data to posteriorly join them into groups. The 11 clusters (upper right) were calculated from the inference of optimal cluster number analysis. Ploidy number (lower right) is represented by distinct colors.

involving chromosomes 1, 10, and 18 indicating transloca-tions of 1p22 (GRCh37 chr1:87,337,015) to 10p14 (GRCh37 chr10:36,119,061), and from 10p11 (GRCh37 chr10:7,634,373) to 18p11 (GRCh37 chr18:9,868,810). In fact, exact breakpoints could be mapped to base-pair level resolution using reads that span the junction boundary (shown in detail in Supplementary Fig. 7). These events were observed in *Group A* and *Group D*, but not *Group B* or *Group C*. Examination of the bulk sequencing sample found translocations in a subset of reads for the EBI, GSC and Illumina samples, but not the TGen sample. Considering the location of the copy number breakpoints it is likely derived from an abnormal chromosome 18 containing portions of chromo-some 1p and 10p, designated as (der18)(1pter->p22::10p14->10p11::18p11->18q). This abnormal chromosome 18 then is lost in both *Group B* and *Group C*.

**LOH analysis provides insight towards mechanism.** Loss of Heterozygosity (LOH) analysis was conducted for each group-level single-cell sequencing BAM by examining the allele frac-tions of germline SNPs known to be heterozygous within the COLO829 lymphoblastoid cell-lines. LOH provides information about the ratio of the parent of origin for copy number events, and importantly, is an independent analysis not leveraged by the original clustering analysis. To ensure higher quality variants, we used quality-filtered heterozygous SNPs known to have a population-based minor allele frequency above 1%. The LOH analysis leverages heterozygous inherited SNPs, in order to infer loss of paternal/maternal chromosomes, such as would be the case for a shift of an SNP allele fraction from ~0% or 100%. For this analysis, we also compared all combinations of the four main groups (*Groups: A, B, C & D*) to four bulk sequencing runs from Craig et al. (TGen, GSC, Illumina and EBI bulk sequencing of samples) to see if hallmark events visible in the bulk sequencing of prior studies were historical events shared by the single-cell data, or whether we were identifying newly emerging events. It should be noted that the $\log_2$ fold change is used for

Fig. 4 since the absolute copy number is difficult to obtain in bulk sequencing data and we wished to compare single-cell/bulk sequencing data together. For the bulk and single-cell data in Fig. 4, the $\log_2$ fold change is in reference to the COLO829-BL bulk sequencing data. By comparison, Fig. 3 contains the absolute copy number for the single-cell data, enabled by the ability of the Cell Ranger DNA Pipeline to determine absolute copy number by leveraging the quantum nature of single-cell copy-number analysis.

The Fig. 4 and Supplementary Figs. 8 and 9, the allele fraction for each SNP is plotted within the COLO829 DAPC-defined groups and independent samples of the cell-line. As expected, groups with fewer cells have greater noise in their allele fraction. However, by taking the average of multiple SNPs across a segment of each sub-clone with specific patterns of LOH become evident. For example, the p-arm of chromosome 1 in *Group A* (Fig. 4-ii) exhibits B-allele frequencies of 67/32 consistent with a heterozygous triploid genetic background (Supplementary Fig. 10), while Group C (Fig. 4-i) for the same region is at a copy number of 2 and B-allele frequencies of 98/1.1 suggesting that chromosome 1p is homozygous. Similarly, the analysis of the bulk data from Craig et al. shows that the TGen sample loses an original haplotype whereas the EBI, Illumina, and GSC samples have multiple copies of one of their haplotypes and a copy of the chromosome 1/10/18 translocation (Fig. 4). Notably, we observed that Group B continues to mirror the TGen lineage (Fig. 4-iii).

More broadly, it's evident that the TGen sample has several parallels to one of the single-cell groups identified in this study, both at chromosome 8 and the lack of an abnormal chromosome 18 (der18)(1pter->p22::10p14->10p11::18p11->18q). LOH and log2-fold analysis over these regions, and the lack of anomalous reads at their junction, are consistent with the lack of an abnormal chromosome 18 for the TGen sample (Fig. 4). This suggests a model where certain groups of cells lose the abnormal chromosome 18 during expansion, while others change copy number for chromosome 8. As discussed later, karyotype

**i. Single Cell *Group B:* Lost (der18)(1pter→p22::10p14→10p11::18p11→18q)**

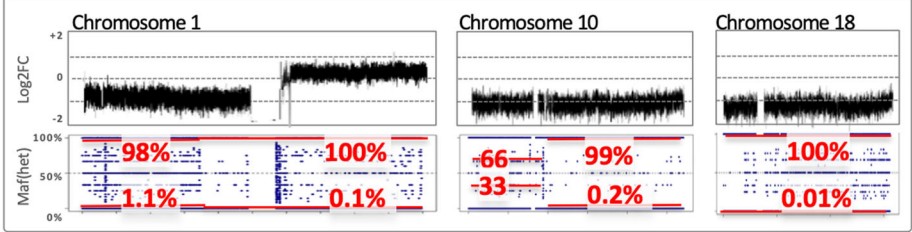
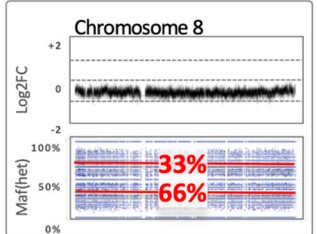

**ii. Single Cell *Group A*: Retained (der18)(1pter→p22::10p14→10p11::18p11→18q)  / + Chr8**

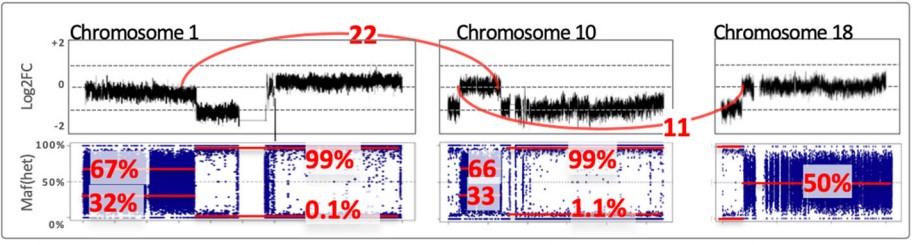
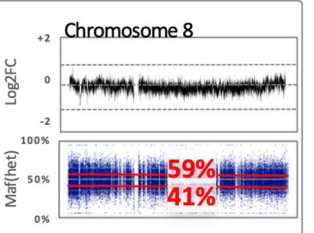

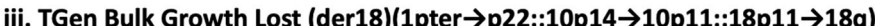

**iii. TGen Bulk Growth Lost (der18)(1pter→p22::10p14→10p11::18p11→18q)**

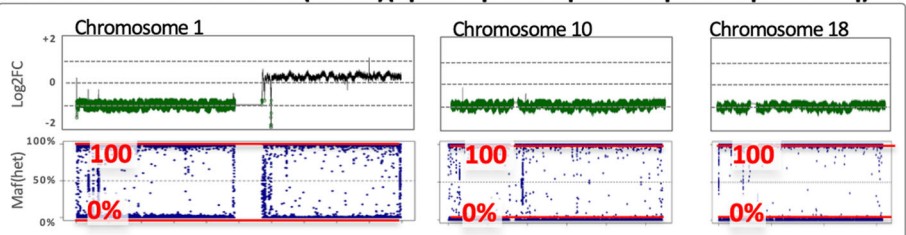
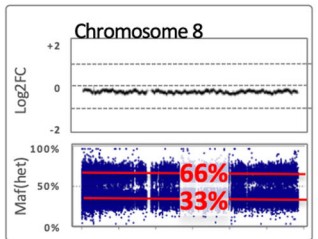

*Craig et al 2016*

**iv. Illumina Bulk: Retained (der18)(1pter→p22::10p14→10p11::18p11→18q)**

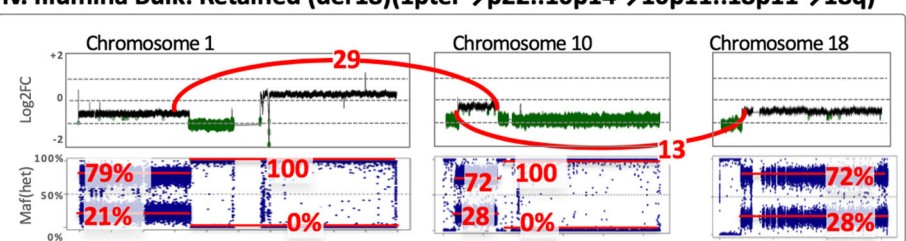
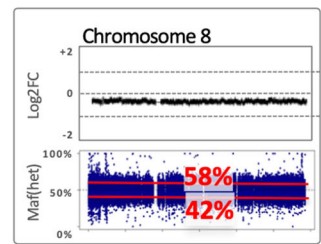

*Craig et al 2016*

**V. Clonality Model**

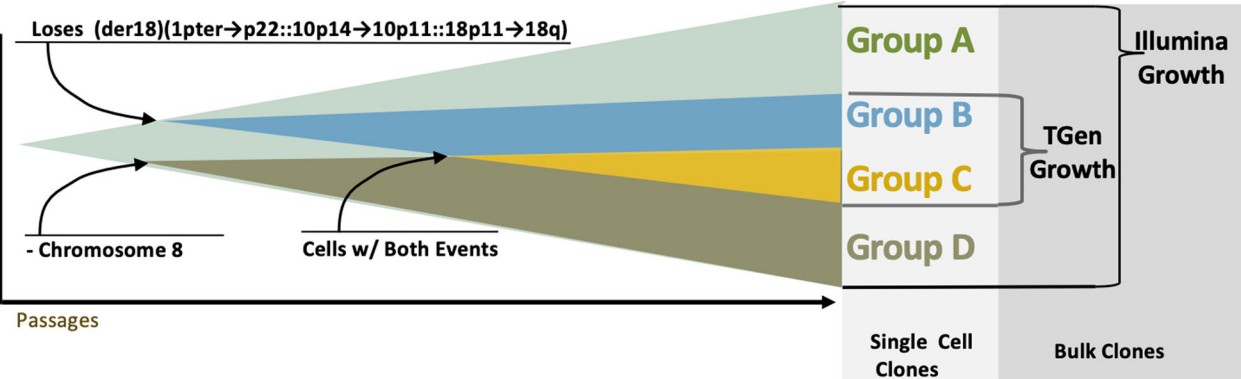

evolution with endoreduplication suggests this would be trisomic, and thus our model reflects a gain of chromosome 8.

**Key events validate spectral karyotyping on independent samples.** We compared the major features determined from single-cell DNA analysis with previously generated karyotyping, of an independent sample, observing remarkable agreement

(Supplementary Fig. 10). All the major rearrangements seen in the karyotype correspond to copy number changes and LOH in the single-cell analysis. The der(?)t(1p?;18q?) of the karyotype is consistent with the (der18)(1pter->p22::10p14->10p11::18p11->18q) postulated from sequencing. The der(?)t(1;3)(q?;p22-24?) x 2 exactly fits the four copies of 1q and most of chromosome 3 up to 3p2. The iso(4q) matches the two copies identified in single-cell

**Fig. 4 Log$_2$Fold$^{ci}$ (upper in each) and Het SNP allele frequency (lower blue in each) for (i) Group B and (ii) Group A, (iii) Illumina Bulk Sample, and (iv) TGen Bulk Sample.** The upper graph of each panel provides an estimated log2 fold change (noting bulk copy number does not inherently produce copy number estimates). For the chromosome 1, 10, and 18 via (p22.3;10p14) and t(p11.22;18p11.22), counts of anomalous reads supporting the junction are shown in red, whereas this event is absent in Group B and C, as well as the TGen Sample. The lower plots of each panel are the allele fraction of known heterozygous SNPs (identified from previous VCFs in the germline lymphocyte lines) for COLO829. Their deviation from the expected 50% allele fraction provides an indication of Loss of Heterozygosity (LOH), where the relative noise is dependent on the number of reads over an SNP and greater spread is observed in groups with fewer reads. The median major/minor allele fractions are provided for each region in red. (v) A Schematic model of the major clones shows a simple model whereby the sub-clones emerge as some cells do not maintain the abnormal chr1p-10q-18q line and/or change in chromosome 8 copy number.

samples. The two copies of der(7)t(7;15)—most of chromosome 7 with small piece of 15 attached to distal 7q—matches the copy number 2 of the tip of 7q (presumably with LOH of the tip) and four copies of 15qter, suggesting that it is der(7)t(7q[near end];15q[near ter]), formed before duplication. The six copies of the region of 7q adjacent to the break is most likely duplication of this region in the translocated copies. The four metaphases where one eleven is replaced by a der(?)t(11;18), may correspond to a few cells that have lost the 1;10;18 and that have a translocated 11 broken mid 11q. The del(6), del(9), and del(16) are consistent with the copy number losses seen on these chromosomes. The del(9) and del(16) are present in 1 or 2 copies, thus probably formed before duplication and some cells losing one copy. For the del(6), the single-cell sequencing detected three copies of normal 6, apparently with allele ratios 1:2. This suggests that the deletion occurred after duplication, and some cells subsequently lost a normal 6, as in the karyotype. Overall, combining the single-cell sequencing with the karyotype enabled us to construct a plausible evolutionary history of the line. Both the karyotype and single-cell sequencing suggest that the cell line duplicated its entire karyotype after most of the rearrangements seen. The translocations—the 1;3, 1;10;18 and 7;15 translocations—fit the Dutrillaux monosomic pattern of karyotype evolution, in which two normal chromosomes are replaced by one unbalanced translocation, resulting in copy number loss and (if before genome duplication) LOH[6,11].

## Discussion

In this study, we performed shallow single-cell sequencing of genomic DNA across 1475 cells from a well-studied cell-line, COLO829, showing that it is fact a mixture of clearly defined sub-clones. These major sub-clones further enabled LOH and breakpoint analysis, providing a clearer picture of clonal heterogeneity, and underlying biology driving the sub-clonal evolution.

Numerous sequencing studies have utilized COLO829 as a reference and benchmark resource, and a few have indicated some evidence for underlying heterogeneity of this line. In this study, we have identified sub-clones with unique hallmark features that provide a potential explanation of previously reported variability in the samples of COLO829, and single-cell methods provide clear insight into the underlying diversity of the cell-line. Specifically, we observe an extra copy of a suggestive break on chromosome 18 that is consistently maintained in some daughter cells. Similarly, we observe that in some cells there is also a change in the copy number of chromosome 8. In addition, it should be remarked that SKY data presented here were available via an online resource and played an essential role in validating the initial findings by single-cell copy number.

A key aspect of the single-cell sequencing of DNA was the efficiency and accuracy of the clustering analysis to identify clonal groups of cells, enabling downstream analysis leveraging bulk analysis tools. Following clustering into clones—by DAPC analysis, which relies on K-means and model selection to infer genetic clusters—, we applied tools specific to tumor or clone

analysis, such as LOH and breakpoint analysis, to find events unique to each clone. Without the ability to place cells accurately in a clone, these later searches would have been, at best, unreliable. While not necessary for our purposes, it would also have been straightforward to identify sequence-level mutations specific to a clone, which would be useful in patient samples.

It is clear that much further algorithm development is possible. For example, se show how LOH analysis enables characterizing clones, and tools provided by other groups, such as SCOPE, CopyMix, and CCNMF, may also provide a new window into single-cell somatic mosaicism[12–14].

In general, despite numerous papers identifying some aspects of the sub-clonal heterogeneity within this cell-line, they gave a fragmented view because they could not assign structural and mutational events to clearly defined clones. Here, we observe a remarkable agreement between the single-cell sequencing, SKY karyotype analysis, and detailed copy number changes.

Our analysis shows how the COLO829 cell line is evolving. The same hallmark features of sub-clones are evident in multiple samples and multiple technologies, indicating that the same sub-clones are present across samples, and that the samples capture different stages in the evolution of the line. COLO829 exhibits a classical pattern of karyotype evolution with endoreduplication described first by Dutrillaux and colleages[15–18]. Two major patterns of karyotype evolution are 'trisomic' and 'monosomic': trisomic is a tendency to gain whole chromosomes, while monosomic is a tendency to form unbalanced translocations with net loss of a chromosome. For example, the emergence of sub-clones replacing a copy of chromosome 18 by (der18)(1pter → p22::10p14 → 10p11::18p11 → 18q) is consistent with monosomic karyotype evolution. In contrast, the larger gain of chromosome 8 follows the trisomic pattern, since LOH analysis suggests a gain rather than loss of chromosome 8. Such chromosomal events defining sub-clones are well studied historically[19–22], but have been largely ignored in the era of bulk sequencing. With the expansion of single-cell methods that have been transforming RNA and DNA sequencing, we see a 'back to the future' opportunity to return to these biology principles with both high-throughput and high resolution.

## Methods

**Preparation of the single-cell suspension.** COLO829 cell line was obtained from American Type Culture Collection (ATCC), Manassas, VA. Cells were cultured in their recommended media conditions at 37 °C. Prior to FACS sorting, the cancer cell line was trypsinized, followed by inactivation with FBS and washed by centrifugation at $300 \times g$ in 1× PBS with 0.04% BSA. Cells were counted and resuspended in recommended media at a final concentration of 1e6/mL in a FACS tube. Two microliters of Vybrant® DyeCycle™ Green stain was added to the cell suspension and incubated at 37 °C for 30 min in the dark. Cells were then analyzed and sorted on a flow cytometer using 488 nm excitation and green emission gating on cells in the G1 phase of the cell cycle (Supplementary Fig. 1). Cells were counted post sorting to ensure accurate concentration.

**Single-cell DNA library generation.** The single-cell suspension was processed using chromium single-cell CNV solution (10× Genomics) as described in the user guide to generating a barcoded single-cell DNA library (Fig. 1a). Single-cells were

partitioned in a hydrogel matrix by combining with a CB polymer to form cell beads (CBs) using a microfluidic chip. Post the first encapsulation, CBs are treated to lyse the encapsulated cells and denature the genomic DNA (gDNA). The denatured gDNA in the CB is then accessible to amplification and barcoding. A second microfluidic encapsulation step is required to partition the CB with 10× barcode gel beads (GBs) to generate an emulsion called GEMs. Immediately after barcoding and amplification, 10× barcoded fragments were pooled and attached to standard Illumina adaptors. Finally, sequencing libraries were quantified by qPCR before sequencing on the Illumina platform using NovaSeq S4 chemistry with 2 × 100 paired-end reads (Fig. 1b).

**Single-cell CNV calling using Cell Ranger DNA**. Paired-end reads were processed using version 1.0 of the Cell Ranger DNA Pipeline (10× Genomics)[8,9]. As described previously, the pipeline consists of barcode processing, alignment to the (hg19) genome and the identification of cell-associated barcodes. Copy number calling is performed on each barcode separately after masking out regions of the genome with low mappability and normalizing for GC content. 1475 barcodes were defined as cells, roughly all barcodes with greater than 1/10th the number of reads as the maximum per-barcode read count. Cells flagged as noisy by the pipeline (102 cells, 6.9%) were removed from the downstream analysis, leaving behind 1373 cells. Cumulative breath of coverage were explored using different depths to calculate coverage across the genome (Supplementary Fig. 11).

**Clustering of single-cell CNV data**. Single-cell CNV calls were extracted from a BED file generated by the Cell Ranger DNA Pipeline for 1373 cell barcodes. Events were filtered to include those with a size ≤ 2Mbp and with a confidence score > 15. Events from different single-cells were grouped together if they had identical copy number and shared 90% reciprocal overlap. Next, events present in less than 5% of cancer cells were discarded. This analysis generated a binary CNV mutation matrix with 112 polymorphic events ranging in size from 2.1 to 147.6 Mbp. The custom R script used to perform this analysis is included as supplementary codes 1 and 2.

To identify clusters we implemented the fast maximum-likelihood (ML) genetic clustering and Bayesian Information Criterion (BIC), subsequently, using the Bioconductor adegenet package (version 2.1.1). BIC curve suggested between 10 to 15 clusters, where 11 ($k = 11$) were selected as the optimal clustering solution (Fig. 2a, b). This yielded four major groups which were explored for sequencing data quality purposes by group (Supplementary Table 1) and per cell (Supplementary Data 1). This four major groups showed a sub-structure within two of them (Fig. 2c). A list of barcodes per major group (Supplementary Data 2) and CNV events across the 11 clusters and 4 groups (Supplementary Data 3) were generated.

**Bulk copy number and loss of heterozygosity analysis**. A python script was used to split the BAM file by barcode assignment, generating a BAM file for each sub-clone (Supplementary Code 3). Cell Ranger DNA Pipeline version 1.1 this functionality is a new sub-pipeline. The new BAMs only include reads from barcodes that are assigned to that cluster to enable downstream analysis with traditional mutation calling tools developed for bulk-data.

Fold change and loss of heterozygosity (LOH) analysis was performed using previously developed tools, tCoNuT(1.0) (https://github.com/tgen/tCoNuT) and DNACopy (version 1.48.0) for both the single-cell grouped BAMs, and for previous bulk sequencing of COLO829 samples[2]. The previous bulk sequencing was conducted by four groups with independent samples of both COLO829 melanoma line and the paired COLO829BL germline lymphoblastic cell-line. Consistent with prior publications, these are referred to as the TGen, EBI, GSC, and Illumina samples. The use of additional copy number analysis tools provided a framework for comparing aggregated sub-clone data to previous bulk sequencing and added additional analysis capabilities such as loss-of-heterozygosity.

LOH was examined using germline heterozygous SNPs for COLO829 using the companion COLO829BL. Specifically, heterozygous germline variants were identified using GATK's HaplotypeCaller (version 3.5.0) from previous sequencing data[23]. The VCFs were annotated with dbSNP 147 using snpEff (version 3.5 h) and input to the tCoNuT parseMergeVCF script (1.0) to output heterozygous SNPs. Heterozygous SNPs were identified along with reads supporting alternative and reference alleles. Only SNPs within a range of 0.4–0.6 allele fractions were utilized from germline 80x whole-genome sequencing data. Whereas tCoNuT converts to an absolute minor allele fraction, figures are shown to span allele fractions from 0 to 1, and both minor and major allele fractions are provided in figures. Using allele fractions, variants near 0.5 still retain alleles from both the maternal and paternal haplotypes, whereas as those nearing 0 and 1 have lost this heterozygosity. Intermediate levels can often be interpreted across a range, such as 0.66/0.33 allele fractions with copy number of 3. The median allele fractions for the minor/major allele were obtained across copy number segments within the single-cell CNV bed file. These regional LOH values are shown in Fig. 4 with red text for the region spanned by the red line.

**Variant detection and breakpoint analysis**. For this analysis, we utilized a previously validated script for the detection of anomalous read pairs (tgen_somaticSV) to identify clusters of read-pair mappings consistent with translocations, inversions, and other structural variants (https://github.com/davcraig75/tgen_somaticSV)[18]. The tool defines donor and acceptor regions and counts read-pairs supporting each, where each donor/acceptor region spans no more than 3× the insert distance and is greater than 10,000 bp in separation. A reference set of reads is required, similar to a tumor/normal set, and for this analysis, we utilized the other groups or the COLO829BL reference line. As the key events were in two of the groups, the former method did not yield meaningful results, and the key region was identified in comparison to references. Additional filters included requiring reads supporting both directions, e.g., determined by first read and mapping quality >20. Anomalous paired reads clusters were sorted by a number of reads, and examined for reads within two megabases of a CNV breakpoint. In some cases, more than one read cluster was evident and in those cases, we prioritized those nearest to change in copy number.

We attempted to identify point mutations specific to clones. Briefly, we called unique variants per major group by systematically comparing four groups using Strelka2 small variant caller and filtering the output using the following criteria: Filter=PASS and QSS_NT > 30. In addition, we calculated the transition transversion ratio and visually examined variants within IGV, filtering out variants known to be present. Examination of coding variants did not yield any high-quality novel point mutations specific any clone and not within the original COLO829 line. The lack of novel mutations in this hyper-mutated line might be because it is a cell-line, rather than each clone resulting in clonal evolution as in the case of patient-derived tumors where the following methods have been found effective.

**Statistics and reproducibility**. Statistical analyses were performed using, mainly, R software environment for statistical computing and Python on High Performance Computing. The packages used in these software are mentioned through the text in the methods section.

**Reporting summary**. Further information on research design is available in the Nature Research Reporting Summary linked to this article.

## Data availability

Data have been deposited in the following 10X Genomics hosted link: https://support.10xgenomics.com/single-cell-dna/datasets/1.0.0/colo829_G1_1k and in GEO; under accession number GSE151409. In addition, Dr. Velazquez-Villarreal will be responsible for replying for any relevant data required upon request.

## Code availability

Specific R and Python code is available in Supplementary Information as Supplementary Code 1–3. These files include comments of the described statistical steps and parameters.

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

## Acknowledgements

This work was supported by 10× Genomics and the Department of Translational Genomics, Norris Comprehensive Cancer Center, and Keck School of Medicine of University of Southern California.

## Author contributions

E.I.V., S.M. D.W.C., J.S., and C.C. contributed to the experimental design. C.C. and Y.Y. conducted the experiments. S.M., E.I.V., D.W.C, M.W., V.K., and Y.Y. contributed to the data analysis. M.G. and P.A.E. contributed to the Spectral Karyotype analysis. E.I.V. did the writing. E.I.V., S.M., J.S., I.T.F., V.K., Y.Y., M.W., C.C., J.D.C., and D.W.C. oversaw all aspects of the manuscript.

## Competing interests

The following authors were employees of 10X Genomics while engaged in the research project: S.M., J.S., I.T.F., V.K., Y.Y., and C.C. The rest of the authors do not have competing interests: E.I.V., J.D.C., D.W.C., M.G., P.A.E., and M.W.
