## [Peer Review File · Communications Biology]

Reviewers' comments:

Reviewer #1 (Remarks to the Author):

Dear Editors,

I went over the manuscript titled "Resolving sub-clonal heterogeneity within cell-line growths by single cell sequencing genomic DNA" by Velazquez-Villarreal et al.

In their manuscript, the authors performed copy-number variation analysis on thousands of individual cells using a novel single cell technology in order to characterize clonal heterogeneity in a melanoma cell line.

The authors first found clusters of cells with similar DNA copy-number profiles and then grouped the reads from all cells within each cluster and created an "in-silico" bulk samples representing the major clones. Then, they performed breakpoint analysis on these bulk samples, which was supported by LOH analysis that was conducted by examining the allele fractions of germline SNPs known to be heterozygous within this cell line. Finally, they validated their results using spectral karyotyping.

I find the paper novel and interesting and would really like to see it published. The results are, as far as I can tell, correct.

However, I would like the authors to consider the following comments in order to make their paper more understandable:

1. Page 6 and Fig. 2C: "Plotting the single cells on the coordinates of two primary DAPC axes revealed four distinct groups (A-D) of cells (Fig. 2C), with sub-structure within two of them (Fig. 2C)." - It will be useful to label the four distinct groups in Fig. 2C (maybe I missed something but I can see them there)
2. Fig. 2C – the cluster numbers overlap and some of them cannot be seen (e.g. clusters no. 4 and 8,9,10).
3. Fig. 3 – I recommend labelling the top colorbar as "cluster number", in the same way that the bottom colorbar was labeled "ploidy"
4. Fig. 3 – it will be helpful if the various ranges referred to in the text (1pter-1p22.3, 10p14-p11.22 etc.) are somehow marked in the figure.
5. Supplementary Fig. 10 - I had a hard time understanding this figure and its legend. I was able to guess what the authors want to show but I couldn't understand it from the figure legend or from the figure itself. I will be helpful to clarify. For example, the sentence "In the right is a split view of chr10p14 and Chr18p11 where on the partner of light green reads on chromosome 10p11(left) map to Chr18p11.32 (pink reads)" doesn't make sense to me.
6. Page 9, 1st paragraph – The text in this paragraph was very confusing to me. The text refers to Fig. 4 and supplementary Fig. 8 and discusses allele frequencies in groups A and B. However, in the figures referred to I could not find any panel depicting Group B. Same for EBI and GSC. I am not an expert in this type of analysis, but I urge the authors to check if the figures and text are consistent here.
7. Finally, there are minor errors in English and I would recommend that an English editor will go over the manuscript and fix them.

Reviewer #2 (Remarks to the Author):

Comments for Author

Increasing evidence suggest that some of the widely used cancer cell lines indeed carry genomic heterogeneity usually undetectable with conventional bulk cell genomic analyses. Velazquez-Villarreal et al used a well-established single-cell microfluidic platform to carry out ScCNV analysis of 1475 individual cells of melanoma cell line COLO829. Authors were able to identify four major sub-clones in the COLO829 cell line using single cell copy number data. Interestingly, loss of heterozygosity (LOH) and breakpoint analysis further identified structural variants and chromosomal rearrangements specific to chromosomes 1, 10 and 18 in at least two of the sub-clones.

I expect that the single cell CNV analysis approach taken by Velazquez-Villarreal et al will be adopted by the wider scientific community especially cancer biology field to study cellular heterogeneity with respect to disease aetiology and treatment resistance. This work will also set benchmark for future genomics studies involving other cancer cell lines. Overall, this study highlights the importance of understanding somatic alterations at single cell level critical for understanding sub-clonal evolution of cancer cell population.

Strengths:

- This study provides a major technical and bioinformatics advancement in the field of single cell genomics
- Authors were able to do LOH and breakpoint analyses for each CNV group using single-cell sequencing BAM files.
- Their single cell data also recapitulated some of the sub-clone hallmark features reported in previous bulk sequencing and spectral karyotyping studies

Weaknesses:

- Only one cancer cell line from one time point was chosen for this study. However, to make it a valuable resource for the field and increase its impact, it is highly recommended that additional melanoma cancer cell lines (or cell lines representing other cancer streams) are also included in this study.
- The smallest CNV group identified in this study represents 43 cells. This highlights the sensitivity of the single cell CNV method but on the other hand, with only 1475 cells analysed, it is possible that some of the rare sub-clones were missed in the analyses.
- Further experiments involving integration of the current dataset with single cell transcriptome profile of COLO829 cells is recommended. Such integrated dataset will provide deeper molecular and functional insights into sub-clonal heterogeneity of the tested cell line.
- A small number of references are cited in the manuscript, especially in the introduction and discussion sections. The discussions section needs a robust discussion of the results.
- Figure panels are not referred while discussing results. It is left to the reader to correlate text with appropriate result panels e.g Figure 4 (panels i-v)
- Included supplementary Figures 7 and 11 are not referred/mentioned in the main text.
- Supplementary Figure 7 legend – Upper graph and lower graph seems to be in reverse order.
- Chromosome numbers on Supplementary Figure 9 are not clearly visible.
- My understanding is that the cartoon depicted in figure 1 (top panel) is largely inspired from the 10X Chromium website and associated online sources. It is desirable that authors use original and distinct cartoon and colour scheme for this manuscript.

I hope authors will be able to address the issues raised above and revise the manuscript accordingly.

Reviewer #3 (Remarks to the Author):

Velazquez-Villarreal et al. performed shallow single cell sequencing of genomic DNA across 1475 cells from the same cell line and identified four major subclonal groups based on CNV analysis. They compared all combinations of the four main groups to four different bulk sequencing runs of the cell line to see whether the CNV subgroups identified by this analysis were previously identified by bulk sequencing or whether these groups represent newly emerging events. The authors inferred a possible evolutionary history for the cell line by combining single cell sequencing with karyotype and demonstrate how bulk sequencing can be misleading in subclone identification.

This project is a nice illustration of how the 10x single cell CNV workflow including the Cell Ranger DNA pipeline can be utilised to study genetic heterogeneity within cancer cell lines. The use of a cell line that has been previously extensively profiled by bulk techniques to clarify subclonality within the cell line works well as a proof of principle. The paper presents a fairly straightforward utilisation of CNV identification at single cell resolution. While perhaps not essential, the work could be extended both to gain greater biological insight and/or from a technical perspective.

A few ideas of possible extensions of the work that would strengthen it could include:

1) Comparison of single cell CNV analysis from the same cell line taken from multiple sources (as is done with bulk sequencing data) to examine variation in subclone composition to further illustrate the genetic heterogeneity between the same cell line grown in different places.

2) Extending the analysis to include one or more additional previously well-characterised melanoma cell lines.

3) Comparison of CNV determination by Cell Ranger DNA pipeline with another CNV determination method for single cell DNA-Seq to demonstrate the consistency and accuracy of the results (i.e.. SCOPE, see Wang et al. 2019, BioRxiv <https://doi.org/10.1101/594267>).

4) While perhaps beyond the scope of this work, scRNA-Seq of COLO829 in parallel with scDNA-Seq would be interesting, both as an attempt to link expression heterogeneity with genetic heterogeneity and because a comparison of CNV inference from the scRNA-Seq data (using HoneyBADGER, inferCNV or something similar) and the scDNA-Seq data could be of interest to scRNA-Seq users attempting to infer CNVs from single cell gene expression data. Additionally, it could serve as another angle for demonstrating what is lost by bulk analysis (in the case of scRNA-Seq, bulk expression analysis).

5) From a biological perspective, it would be interesting to extend the discussion to address how models of subclonal diversity can be leveraged to understand cancer evolution/address the potential clinical significance.

A couple very minor comments to improve readability - 1) It would be better if Figure numbering corresponded to the order in which figures are initially mentioned/referred to in the main text. 2) Figure 2A-B - font size of axes should be increased.

Overview

We are excited to submit this paper and we hope you believe it to be of interest to your readers. We include bulk-sequencing, single-cell sequencing, spectral karyotyping to synthesize data spanning generated by multiple institutions, using independent technologies/methodologies and over multiple years. The end result is a cohesive article that demonstrates the power to uncover important biological feature - while well established in the literature - were not part of the current understanding of this cell-line, despite extensive studies.

All three reviewer responses were positive on the article's merits and are encouraging of its publication within *Communication Biology*.

Reviewer 1 writes:

"I find the paper novel and interesting and would really like to see it published".

Reviewer 2 similarly highlights the importance:

"This study provides a major technical and bioinformatics advancement in the field of single cell genomics".

Reviewer 3 is also positive

"This project is a nice illustration of how the 10x single cell CNV workflow including the CellRanger DNA pipeline can be utilised to study genetic heterogeneity within cancer cell lines."

Unfortunately, our initial presentation of the work led to a suggestion of adding additional cell-lines by two of the reviewers, as emphasized by the editor. This was largely due to an incomplete background section that did not appropriately show how the features we identified within this well studied cell-line are based on well-established biology. In fact, the paper contains genetic analysis by 3 different technologies (bulk, single-cell, SKY), by over 5 groups, and more than 5 growths of COLO-829. It's important to consider the request of the context of additional cell-line data.

Reviewer 2 wrote:

However, to make it a valuable resource for the field and increase its impact, it is highly recommended that additional melanoma cancer cell lines (or cell lines representing other cancer streams) are also included in this study.

Reviewer 3 similarly writes:

Extending the analysis to include one or more additional previously well-characterized melanoma cell lines....

First, we believe that adding additional cell-lines would fundamentally change the paper.

This paper is focused on the biology of a tumor/normal cell-line that is unparalleled in its study and builds using data from multiple resources. In the revised version, we now emphasize that COLO829 exhibits a classical Dutrillaux's monosomic pattern of karyotype evolution – a biological phenomenon that is well established and well-studied over the past 25 years¹, found in colorectal, breast, and melanoma lines. Additional cell-lines would take away from the completeness of the extensive amount of data for COLO-829, essentially morphing this focused story into a survey paper – rather than showing elucidation of the biology.

We believe *Communication Biology* is the right journal as “we present significant advances bringing new biological insight to a specialized area of research” and have stayed focus on biology rather than crafting a survey paper, consistent with the journal goals.

¹ Muleris M1, Dutrillaux B. The accumulation and occurrence of clonal and unstable rearrangements are independent in colorectal cancer cells. *Cancer Genet Cytogenet.* 1996 Nov;92(1):11-3.

Second, there is not really comparable line or set of lines to add, without changing all aspects of the paper. This paper is largely a deep-dive and not a survey paper. The time, effort, and cost required to bring additional lines and data to a similar level are beyond the time-frame of review and beyond the means of the authors, for an effort to show biology in the literature for 25+ years. Growing, sequencing, and validating multiple cell-lines is simply beyond the scope of this paper which already introduces novel methods, and integrates COLO-829 data from multiple groups, prior publications, and included spectral karyotyping. Accordingly, in the revised version, we highlight the extensive amount of literature and research on COLO-829. This additionally addresses a later critique of having limited references.

Third, the request for more cell-lines does not address a biological critique, rather probably emphasizes an underlying enthusiasm for the insight gained by this method. This is emphasized by Reviewer 3, who present their feedback as an idea, but not as a major critique”, noting they specifically write:

A few ideas of possible extensions of the work that would strengthen it could includes Comparison of single cell CNV analysis from the same cell line taken from multiple sources (as is done with bulk sequencing data) to examine variation in subclone composition to further illustrate the genetic heterogeneity between the same cell line grown in different places.

We considered this request and logistics extensively, simply by the logic of “*the reviewers suggested it*”. In the end, the request for surveying additional lines is a significant and massive undertaking that undermines the “deep dive” taken here. The reviews are generally positive, and indeed the request reflects the enthusiasm and why we are excited to publish this article in *Communication Biology*.

We recognize the request for more cell-lines had solid foundation built on the text we wrote. We have revised the text still to address this critique, emphasizing that multiple growths for COLO-829 were part of an earlier publication. Specifically, Craig *et al* (2016, *Scientific Reports*) coordinated such an effort over multiple years and indeed concluded with peer review that growths showed variability. We emphasize that it has been previously shown that repeating culturing will lead to differences in clonal evolution. Showing it again - yet at the single-cell level is minimally 6 to 12 months, yet simply to prove what is previously established. Second, we also provide more review of classical Dutrillaux’s monosomic pattern of karyotype evolution as mentioned earlier. Indeed, the fact that the data within this run gave so much insight into mechanism is a testament to the utility and impact of these approaches. The revised paper then remains focused deep dive on a well-established line, integrating in multiple technologies by multiple groups using multiple growths.

In summary, we definitely appreciate the rationale behind the request for more cells – and we believe it speaks towards the enthusiasm we have to these new methods. The reviewers on the whole are very positive. However, we must emphasize that additional cell-lines (1) would detract from the key messages of the paper, (2) comparably studied tumor/normal sets of cell-lines to COLO-829² do not exist; and (3) does not address a clear critique towards biology. Moreover, the paper includes multiple growths already – via the SKY karyotyping and bulk sequencing – which are in agreement with the single-cell sequencing data. Taken together, we think the article will be of considerable interest to *Communication Biology*, and hope the Editors remain enthusiastic

² We are aware of a well-studied Breast Cancer line that is currently part of a separate publication by different authors in review in Nature Biotechnology.

towards the paper that is a deep dive into the biology of a well-established standard, rather than a survey paper.

Major changes include revisions to the introduction and discussion. Specifically, we provide more insight into the model for COLO829. For chromosome 8, we more accurately describe the event as being likely a gain of chromosome 8, recognizing the ambiguity from LOH analysis alone. This is reflected in Figure 4V. In doing so, we have revised the model to indicate where our model assumes from other resources vs. determines from the existing data. Major additions to discussion include the final paragraph:

Our results on COLO-829 lead to several immediate questions, particularly towards the generalizability of our biological observations. The fact that multiple growths and multiple technologies show the same hallmark features within sub-clones does indicate the emergence of sub-clones is reproducible across growths. Indeed, COLO829 exhibits a classical Dutrillaux's of karyotype evolution with endoreduplication¹⁵, where consistent and reproducible sub-clones emerge from the loss of abnormal chromosomes within some COLO-829 cells. Described first by Dutrillaux and colleagues, karyotype evolution that is either 'trisomic' and 'monosomic' leads to a tendency to gain or lose a chromosome respectively. Indeed, the emergence of sub-clones lacking chromosome 18 (der18)(1pter→p22::10p14→10p11::18p11→18q) is consistent with monosomic karyotype evolution while the gain in chromosome 8 is consistent with a trisomic evolution. Unlike for chromosome 18, the latter gain of chromosome 8 in our model relies on Dutrillaux's model in fact since we cannot determine directly from LOH analysis alone whether the end-result reflects a gain or loss of chromosome 8. Indeed, these types of chromosomal events driving the sub-clones at the single-cell are well studied and characterized historically ^{16,17}. Until now, they have been largely ignored in the era of bulk-sequencing. With the expansion of single-cell methods that have been transforming RNA to DNA, we see a 'back to the future' opportunity to return to these biology principles with both high-throughput and high resolution.

Referee #1 Responses

Referee #1: I went over the manuscript titled "Resolving sub-clonal heterogeneity within cell-line growths by single cell sequencing genomic DNA" by Velazquez-Villarreal et al.

In their manuscript, the authors performed copy-number variation analysis on thousands of individual cells using a novel single cell technology in order to characterize clonal heterogeneity in a melanoma cell line.

The authors first found clusters of cells with similar DNA copy-number profiles and then grouped the reads from all cells within each cluster and created an "in-silico" bulk samples representing the major clones. Then, they performed breakpoint analysis on these bulk samples, which was supported by LOH analysis that was conducted by examining the allele fractions of germline SNPs known to be heterozygous within this cell line. Finally, they validated their results using spectral karyotyping.

I find the paper novel and interesting and would really like to see it published. The results are, as far as I can tell, correct.

We appreciate that referee #1 clearly identified novelty and broad interest in the paper. We too wish to have it published, and considerable effort went into the development of the informatic methods. More importantly, we were able to bring in considerable additional data including bulk sequencing and spectral karyotyping. The focus on one cell-line truly shows the insight that can be gained through single-cell CNV analysis.

However, I would like the authors to consider the following comments in order to make their paper more understandable:

1. Page 6 and Fig. 2C: "Plotting the single cells on the coordinates of two primary DAPC axes revealed four distinct groups (A-D) of cells (Fig. 2C), with sub-structure within two of them (Fig. 2C)." - It will be useful to label the four distinct groups in Fig. 2C (maybe I missed something but I can see them there)

These are excellent suggestions and we have made the revisions within the figures to aid in clarity.

2. Fig. 2C – the cluster numbers overlap and some of them cannot be seen (e.g. clusters no. 4 and 8,9,10).

We agree the numbers are confusing. The colors correspond to the coloring within the legend shown in Figure 3, and we have adjusted the legend and figure accordingly.

3. Fig. 3 – I recommend labelling the top colorbar as "cluster number", in the same way that the bottom colorbar was labeled "ploidy"

We have made the recommended change.

4. Fig. 3 – it will be helpful if the various ranges referred to in the text (1pter-1p22.3, 10p14-p11.22 etc.) are somehow marked in the figure.

We made every attempt to put this level of detail within Figure 3. In doing so we reviewed how this is handled in GWAS and many other genome-wide plots. Unfortunately, that level of detail does not render well to precise cytobands, and we have revised the text to refer to the supplemental data. The compromise is to indicate using superscripts for the key regions, and to reference the supplemental data for more detail.

5. Supplementary Fig. 10 - I had a hard time understanding this figure and its legend. I was able to guess what the authors want to show but I couldn't understand it from the figure legend or from the figure itself. I will be helpful to clarify. For example, the sentence "In the right is a split view of chr10p14 and Chr18p11 where on the partner of light green reads on chromosome 10p11(left) map to Chr18p11.32 (pink reads)" doesn't make sense to me.

We have revised this figure's text for clarity. It now reads:

Supplementary Figure 10 | IGV Traces of anomalous reads spanning translocations. Split views for two translocations are shown, Chr1p22 <-> Chr10p11 on the left and Chr10p14 <-> chr18p11 on the right. Read coloring is formally defined within the IGV manual and are set to indicate instances where the insert distances between paired reads is beyond 1kb. We highlight major colors relevant for interpretation. Read with a read-partner mapping within the same chromosome are red, read with their partner mapping to chromosome 18 are in pink, read with a partner mapping to chromosome 10 are blue, read with a partner mapping to chromosome 1 are light green. (Left) Split view of Chr1p22 <-> Chr10p11. Reads in pink are those whose partner maps to chr10p11, whereas reads whose partner maps to chr1p22 are shown in blue. Reads in red read map 2.7Mb upstream. (Right) Split view of Chr10p14 <-> chr18p11. Pink reads indicate where the partner maps to Chr18p11.32. Light green reads indicate where the read partner maps to chromosome 10p11.

The reviewer writes

6. Page 9, 1st paragraph – The text in this paragraph was very confusing to me. The text refers to Fig. 4 and supplementary Fig. 8 and discusses allele frequencies in groups A and B. However, in the figures referred to I could not find any panel depicting Group B. Same for EBI and GSC. I am not an expert in this type of analysis, but I urge the authors to check if the figures and text are consistent here.

The reviewer was correct and valid in their concern. We addressed this in multiple ways: clarifying Figure 4 with improved legends, fixing issues with the text and within the legend of figure 4. Moreover, the text in supplemental figures as well. In part, the text now reads:

To expand on this observation, LOH analysis was conducted by examining the allele fractions of germline SNPs known to be heterozygous within COLO829 lymphoblastic cell-line, which has previously sequenced multiple times. LOH provides information about the ratio of parent of origin for copy number events, and importantly is independent of the original clustering analysis. In order to ensure higher quality variants, we used quality filtered heterozygous SNPs known to have a population-based minor allele frequency above 1% used. Shown in **Fig. 4** and **Supplementary Figs. 7 and 8**, the allele fraction for each SNP is plotted within the COLO829 DAPC-defined groups and independent growths of the cell-line. As expected, groups with fewer cells have greater noise in their allele fraction. However, by taking the average of multiple SNPs across a segment sub-clone specific patterns of LOH become evident.

The Reviewer writes:

7. Finally, there are minor errors in English and I would recommend that an English editor will go over the manuscript and fix them.

We have made a major effort to address these unfortunate errors.

Referee #2 Responses

The reviewer writes:

Increasing evidence suggest that some of the widely used cancer cell lines indeed carry genomic heterogeneity usually undetectable with conventional bulk cell genomic analyses. Velazquez-Villarreal et al used a well-established single-cell microfluidic platform to carry out ScCNV analysis of 1475 individual cells of melanoma cell line COLO829. Authors were able to identify four major sub-clones in the COLO829 cell line using single cell copy number data. Interestingly, loss of heterozygosity (LOH) and breakpoint analysis further identified structural variants and chromosomal rearrangements specific to chromosomes 1, 10 and 18 in at least two of the sub-clones.

I expect that the single cell CNV analysis approach taken by Velazquez-Villarreal et al will be adopted by the wider scientific community especially cancer biology field to study cellular heterogeneity with respect to disease aetiology and treatment resistance. This work will also set benchmark for future genomics studies involving other cancer cell lines. Overall, this study highlights the importance of understanding somatic alterations at single cell level critical for understanding sub-clonal evolution of cancer cell population.

We appreciate that referee #2 highlight the resolution used in our study to detect genomic heterogeneity previously undetected by conventional bulk cell genomic analyses. Importantly, we were able to describe in detail genomic heterogeneity on 1475 single cells from a well-known melanoma cell line COLO829 using well-established single-cell microfluidic platform. More important we were able to identify, previously undetected, structural variants at sub clonal level.

We too expect that our CNV analysis be adopted by the scientific community as a reference for future studies in cancer biology focused on cellular heterogeneity. Most importantly, we understand the relevance of our analysis and its application to study disease etiology and treatment resistance since its higher resolution to detect structural variation. We also expect that our work set a point of reference where future genomic studies can be compared and/or assessed with, especially when involved cancer cell lines. We appreciate how this study was highlighted as an important effort to understand somatic alterations at single cell level, critical resolution needed to detect and understand sub-clonal evolution in cancer cells.

The reviewer writes:

Strengths:

- This study provides a major technical and bioinformatics advancement in the field of single cell genomics
- Authors were able to do LOH and breakpoint analyses for each CNV group using single-cell sequencing BAM files.
- Their single cell data also recapitulated some of the sub-clone hallmark features reported in previous bulk sequencing and spectral karyotyping studies

We appreciate the recognition about how our work provides major advancement in single cell genomics especially related with single cell techniques and bioinformatics which allow us to

perform specialized advanced LOH and breakpoint analyses at sub-clonal level. We completely agree that this is the level of resolution that clarifies past assumptions from bulk experiments where hallmark features were reported using sequencing and karyotyping. We expect too that our study serves as a benchmark to assess hallmark structural variation discoveries in the area of cancer heterogeneity.

The reviewer writes:

Weaknesses:

1- Only one cancer cell line from one time point was chosen for this study. However, to make it a valuable resource for the field and increase its impact, it is highly recommended that additional melanoma cancer cell lines (or cell lines representing other cancer streams) are also included in this study.

Our paper is focused on the insight gained on an extensively well-studied cell-line by multiple groups over many years and genomic technologies. It is not a survey paper. As previously explained, including additional cell-lines fundamentally changes the paper and the validation of such efforts – multiple growths, multiple technologies, and multiple research groups – is beyond the scope.

The reviewer's critique has merit and the question was invoked because of a lack of underlying biological discussion driving the sub-clones. In fact these generalizable events are part of the older literature. In part this is addressed in the final discussion paragraph (as follows in blue), though we have attempted to address the questions behind the request throughout.

Our results on COLO-829 lead to several immediate questions, particularly towards the generalizability of our biological observations. The fact that multiple growths and multiple technologies show the same hallmark features within sub-clones does indicate the emergence of sub-clones is reproducible across growths. Indeed, COLO829 exhibits a classical Dutrillaux's of karyotype evolution with endoreduplication¹⁵, where consistent and reproducible sub-clones emerge from the change in chromosome copy number within some COLO-829 cells. Described first by Dutrillaux and colleagues, karyotype evolution that is either 'trisomic' and 'monosomic' leads to a tendency to gain or lose a chromosome respectively. Indeed, the emergence of sub-clones lacking chromosome 18 (der18)(1pter→p22::10p14→10p11::18p11→18q) is consistent with monosomic karyotype evolution for chromosomes with translocations while the gain in chromosome 8 is consistent with a trisomic evolution. Unlike for chromosome 18, the latter gain of chromosome 8 in our model relies on trisomic model, since LOH analysis does determine whether the end-result reflects a gain or loss of chromosome 8. Indeed, these types of chromosomal events driving the sub-clones at the single-cell are well studied and characterized historically^{16,17}. Until now, they have been largely ignored in the era of bulk-sequencing. With the expansion of single-cell methods that have been transforming RNA to DNA, we see a 'back to the future' opportunity to return to these biology principles with both high-throughput and high resolution.

We have also added to parts of the introduction, including in the first paragraph:

We follow this analysis with a deep-dive examining data by multiple technologies and multiple growths on COLO829 in order to better understand the resulting sub-clonality, their hallmark features, and the underlying driving biology.

Lastly, we reflect this within the abstract:

... Based on clustering, break-point and loss of heterozygosity (LOH) analysis of aggregated data from sub-clones, we identified an abnormal chromosome 18 (der18)(1pter->p22::10p14->10p11::18p11->18q) containing translocations to chromosomes 10 and 1 that was lost in two of four sub-clones. Additionally, two of the sub-clones were distinguished by change in chromosome 8 copy number. The sub-clones hallmark events were validated within bulk sequencing and spectral karyotyping data, each from independent growths and by independent research teams. Overall, COLO-829 exhibits a classical Dutrillaux's monosomic/trisomic pattern of karyotype evolution with endoreduplication, where consistent and reproducible sub-clones emerge from the loss/gain of abnormal chromosomes within some COLO-829 cells. Taken together, our results demonstrate how shallow copy number profiling together with clustering analysis of single cell sequencing can uncover significant hidden insights even in well studied cell-lines.

Overall, we are concerned that more melanoma cancer cell lines or cell lines representing other cancer streams will essentially transform the focused story into a survey paper, diluting the insights reported on this COLO-829 focused study with unvalidated observations.

The reviewer writes:

2- The smallest CNV group identified in this study represents 43 cells. This highlights the sensitivity of the single cell CNV method but on the other hand, with only 1475 cells analysed, it is possible that some of the rare sub-clones were missed in the analyses.

As highlighted, single cell CNV method allows the identification of cells with different cancer heterogeneity, some sub-clones may have distinct characteristics since their active adaptation and evolution. Our clustering process is focused on identify groups with similar characteristics, taking this in context and as it is mentioned, there is always the possibility to regroup rare sub clones in small groups with similar characteristics but less likely to cluster small number of cells of rare sub clones creating new groups; in other words by increasing the number of cells is more likely that rare cells will feed the already identified groups, already detected through our clustering process.

The reviewer writes: 3- Further experiments involving integration of the current dataset with single cell transcriptome profile of COLO829 cells is recommended. Such integrated dataset will provide deeper molecular and functional insights into sub-clonal heterogeneity of the tested cell line.

We emphasize the methodologies for single cell DNA presented here are on their own entirely novel. **We did bring additional methods, such as SKY karyotyping and these indeed provided validation of the single-cell DNA results.**

The requested change of RNA is inherently indirect. We emphasize this paper is focused on the direct interrogation of the somatic DNA landscape, **not the RNA** transcriptomic landscape.

Moreover, it's currently not technologically possible to conduct single-cell sequencing of both RNA and DNA from the same cells. One could sequence RNA and DNA separately and develop methods for integration. These new methods would be needed to infer cells, and that in turn would need validation. While we can imagine such methods for these samples, we are not sure the insight provided to this paper on somatic DNA landscape. By the same argument, we could ATAC-seq – though nothing is really added to our understanding of the DNA landscape.

The reviewer writes: 4 - A small number of references are cited in the manuscript, especially in the introduction and discussion sections. The discussions section needs a robust discussion of the results.

This area was particularly an area of focus, and we believe that the added background and discussion show the novelty and why additional cell-lines is something we are all excited about but really needs to be part of future efforts, building upon these foundations.

These changes are already mentioned.

The reviewer writes: 5 - Figure panels are not referred while discussing results. It is left to the reader to correlate text with appropriate result panels e.g Figure 4 (panels i-v)

We thank the reviewers for finding this oversight. We now include references to Figure 4 in the text while discussing results to make readers easy walk through the text. Examples from the text now read:

For example, the p-arm of chromosome 1 is determined to be at a copy number of 3 in Group A (Fig. 4-ii) and exhibits B-allele frequencies of 67/32 consistent with a heterozygous triploid genetic background (Supplementary Fig. 11), while Group C (Fig. 4-i) for the same region is at a copy number of 2 and B-allele frequencies of 98/1.1 suggesting that chromosome 1p is homozygous diploid for one of the parents. Similar analysis of the data from Craig et al, we identified that TGen growth lost an original haplotype where EBI, Illumina, and GSC growth had multiple copies of one of their ploidies and still a single copy of the chromosome 1 translocation (Fig. 4). Notably, we observed that Group C continues to mirror the TGen lineage. In this example, it is clearly observed evidence from the homozygous deletion in TGen (Fig. 4-iii) where the clustering was biologically correct (Supplementary Fig. 10).

The reviewer writes: 6 - Included supplementary Figures 7 and 11 are not referred/mentioned in the main text.

As suggested, supplementary Figures 7 and 11 were included in the main text.

The reviewer writes: 7 - Supplementary Figure 7 legend – Upper graph and lower graph seems to be in reverse order.

The reviewer has a keen eye, and we have made this correction. Also, there was an issue with a label of Figure 4 Group B that is now corrected.

The reviewer writes: 8 - Chromosome numbers on Supplementary Figure 9 are not clearly visible.

On Supplementary Fig. 9, Chromosome numbers were edited by increasing font size to make them more visible.

The reviewer writes: 9 - My understanding is that the cartoon depicted in figure 1 (top panel) is largely inspired from the 10X Chromium website and associated online sources. It is desirable that authors use original and distinct cartoon and colour scheme for this manuscript.

Co-authors of this manuscript developed cartoon depicted in figure 1 (top panel) which originally shows the sample preparation steps used in our experiments.

Referee #3 Responses

The reviewer writes: Velazquez-Villarreal et al. performed shallow single cell sequencing of genomic DNA across 1475 cells from the same cell line and identified four major subclonal groups based on CNV analysis. They compared all combinations of the four main groups to four different bulk sequencing runs of the cell line to see whether the CNV subgroups identified by this analysis were previously identified by bulk sequencing or whether these groups represent newly emerging events. The authors inferred a possible evolutionary history for the cell line by combining single cell sequencing with karyotype and demonstrate how bulk sequencing can be misleading in subclone identification.

This project is a nice illustration of how the 10x single cell CNV workflow including the CellRanger DNA pipeline can be utilised to study genetic heterogeneity within cancer cell lines. The use of a cell line that has been previously extensively profiled by bulk techniques to clarify subclonality within the cell line works well as a proof of principle. The paper presents a fairly straightforward utilisation of CNV identification at single cell resolution. While perhaps not essential, the work could be extended both to gain greater biological insight and/or from a technical perspective

A few ideas of possible extensions of the work that would strengthen it could include:

1) Comparison of single cell CNV analysis from the same cell line taken from multiple sources (as is done with bulk sequencing data) to examine variation in subclone composition to further illustrate the genetic heterogeneity between the same cell line grown in different places.

Indeed, our manuscript includes re-analysis of bulk sequencing samples from multiple sources of COLO829 from *Craig et al 2016*, in the context of scCNV subgroups. This new study opens the possibility for a wide range of comparisons in order to generate new insights. We agree and hope that the scientific community take our work as a reference for these potential comparisons and bring novel insights to the field of cancer genomics heterogeneity,

The reviewer writes: 2) Extending the analysis to include one or more additional previously well-characterised melanoma cell lines.

As we extensively explained before, our study is focused on the biology of a cell-line COLO-829 that is extremely well studied, showing how demonstrate how shallow copy number profiling together with clustering analysis of single cell sequencing can uncover significant hidden insights even in well studied cell-lines. **We are fundamentally concerned that including other melanoma cell lines will fundamentally change the paper and focus, diminishing the insights reported on this COLO-829 focused study.** Each new line would require SKY karyotyping, validation and coordination across multiple groups, and in the end **lead to a very different survey paper that lacks the focus on communicating biology.**

The reviewer's concern has merit - and we have highlighted in earlier portions of this response where the abstract, introduction, and discussion are modified to emphasize the deep-dive into COLO829.

The reviewer writes: 3) Comparison of CNV determination by CellRanger DNA pipeline with another CNV determination method for single cell DNA-Seq to demonstrate the consistency and accuracy of the results (i.e.. SCOPE, see Wang et al. 2019, BioRxiv <https://doi.org/10.1101/594267>).

SCOPE is likely under the review and the referenced article is within pre-print; its comparison would be premature given that is likely being done within another paper. The authors are very excited about SCOPE and such have referenced it in the paper. The text now contains:

In addition to the tools used here, those by other groups such as SCOPE, CopyMix, and CCNMF, may provide a new window into single-cell somatic mosaicism^{12,13,14}.

The reviewer writes: 4) While perhaps beyond the scope of this work, scRNA-Seq of COLO829 in parallel with scDNA-Seq would be interesting, both as an attempt to link expression heterogeneity with genetic heterogeneity and because a comparison of CNV inference from the scRNA-Seq data (using HoneyBADGER, inferCNV or something similar) and the scDNA-Seq data could be of interest to scRNA-Seq users attempting to infer CNVs from single cell gene expression data. Additionally, it could serve as another angle for demonstrating what is lost by bulk analysis (in the case of scRNA-Seq, bulk expression analysis).

We agree that it would be very exciting and well hugely informative to look at single-cell RNA-seq and single-cell CNV. Some of the authors continually push for this sort of innovation from the same cells, while our authors involved in the technology development emphasize the technical challenges in doing so. That said, we all look forward to making those technical leaps. We share the enthusiasm towards the integration and we agree that while beyond the scope is something to pursue down the road.

The reviewer writes: 5) From a biological perspective, it would be interesting to extend the discussion to address how models of subclonal diversity can be leveraged to understand cancer evolution/address the potential clinical significance.

We believe our paper is in part a reminder that biology has been making insights long-before next-generation sequencing. Spectral Karyotyping had identified many underlying biological principles that were generally not considered (or not possible to consider). Specifically, the text now reads:

Our results on COLO-829 lead to several immediate questions, particularly towards the generalizability of our biological observations. The fact that multiple growths and multiple technologies show the same hallmark features within sub-clones does indicate the emergence of sub-clones is reproducible across growths. Indeed, COLO829 exhibits a classical Dutrillaux's of karyotype evolution with endoreduplication¹⁵, where consistent and reproducible sub-clones emerge from the change in chromosome copy number within some COLO-829 cells. Described first by Dutrillaux and colleagues, karyotype evolution that is either 'trisomic' and 'monosomic' leads to a tendency to gain or lose a chromosome respectively. Indeed, the emergence of sub-clones lacking chromosome 18 (der18)(1pter→p22::10p14→10p11::18p11→18q) is consistent with monosomic karyotype evolution for chromosomes with translocations while the gain in chromosome 8 is consistent with a trisomic evolution. Unlike for chromosome 18, the latter gain of chromosome 8 in our model relies on trisomic model, since LOH analysis does determine whether the end-result reflects a gain or loss of chromosome 8. Indeed, these types of chromosomal events driving the sub-clones at the single-cell are well studied and characterized historically^{16,17}. Until now, they have been largely ignored in the era of bulk-sequencing. With the expansion of single-cell methods that have been transforming RNA to DNA, we see a 'back to the future' opportunity to return to these biology principles with both high-throughput and high resolution.

The reviewer writes:

A couple very minor comments to improve readability –

1) It would be better if Figure numbering corresponded to the order in which figures are initially mentioned/referred to in the main text.

We have made these changes throughout the text.

The reviewer writes:

2) Figure 2A-B - font size of axes should be increased.

We have made the recommended change.

REVIEWERS' COMMENTS:

Reviewer #2 (Remarks to the Author):

In my view, the overall quality of the manuscript has improved as authors have addressed some of the critical issues raised in the last round of review. Specifically, the abstract and the introduction sections now include a clear rationale and a message regarding the use of novel single-cell DNA sequencing workflow to study COLO829 cell line. However, the discussion section still needs some attention. For example, the following sentences are ambiguous:

"A major general and far-reaching observation is also made of how single-cell shallow sequencing is a starting point that enables further iterative analysis of breakpoints and SNPs, and while not shown in this study mutation-specific clones. While in this study we show how LOH analysis enables characterizing clones, it is clear from these studies that much further algorithm development is possible".

Regarding Figure 1(top panel), I will suggest that authors make sure that they are not self-plagiarising their previous published artwork.

All the best to authors.

Reviewer #3 (Remarks to the Author):

I recommend the revised manuscript for publication in Communications Biology. I believe the text of both the introduction and discussion were improved and now give fuller context. The manuscript still needs to be edited for grammar and clarity.

Overview

We are excited to submit this revised version of our paper. We appreciate all your positive comments on our work and your intention in principle to publish a suitable revised version in Communications Biology.

All two reviewer responses were positive on the article's merits and are encouraging of its publication within Communication Biology.

Reviewer 2 writes:

"In my view, the overall quality of the manuscript has improved as authors have addressed some of the critical issues raised in the last round of review. Specifically, the abstract and the introduction sections now include a clear rationale and a message regarding the use of novel single-cell DNA sequencing workflow to study COLO829 cell line."

Reviewer 3 is also positive:

"I recommend the revised manuscript for publication in Communications Biology. I believe the text of both the introduction and discussion were improved and now give fuller context."

Below we address the comments from both reviewers.

Referee #2 Responses

Referee #2: In my view, the overall quality of the manuscript has improved as authors have addressed some of the critical issues raised in the last round of review. Specifically, the abstract and the introduction sections now include a clear rationale and a message regarding the use of novel single-cell DNA sequencing workflow to study COLO829 cell line.

We appreciate that referee #2 clearly mentioning the improvement of the overall quality of the manuscript as some pointed critical issues on last round of review were addressed. We are happy to read how our considerable effort went into the addressing last round of review.

However, the discussion section still needs some attention. For example, the following sentences are ambiguous:

"A major general and far-reaching observation is also made of how single-cell shallow sequencing is a starting point that enables further iterative analysis of breakpoints and SNPs, and while not shown in this study mutation-specific clones. While in this study we show how LOH analysis enables characterizing clones, it is clear from these studies that much further algorithm development is possible".

We thank the reviewers for pointing these apparently ambiguous sentences. These are excellent suggestions. We have revised these for clarity. It now reads:

"A major general observation is about how single-cell shallow sequencing is a starting point that enables further iterative analysis of breakpoints and SNPs such as identifying mutation-specific clones. In this study, we show how LOH analysis enables characterizing clones, thus it is clear that much further algorithm development is possible".

Regarding Figure 1(top panel), I will suggest that authors make sure that they are not self-plagiarising their previous published artwork.

We have made the suggestion, co-authors of this manuscript developed cartoon depicted in figure 1 (top panel) which originally shows the sample preparation steps used in our experiments.

Referee #3 Responses

The reviewer writes:

I recommend the revised manuscript for publication in Communications Biology. I believe the text of both the introduction and discussion were improved and now give fuller context..

We appreciate that referee #2 highlight our success addressing last round of review.

The manuscript still needs to be edited for grammar and clarity.

We have made a major effort to address this suggestion.